# Structure-evolution-designed amorphous oxides for dielectric energy storage

Yahui Yu[1,7], Qing Zhang[2,7], Zhiyu Xu[1,3,7], Weijie Zheng[1], Jibo Xu[1], Zhongnan Xi[4], Lin Zhu[4], Chunyan Ding[1], Yanqiang Cao[5], Chunyan Zheng[1], Yalin Qin[1], Shandong Li ®[3], Aidong Li[4], Di Wu ®[4], Karin M. Rabe[6], Xiaohui Liu ®[2] ✉ & Zheng Wen ®[1,3] ✉

Recently, rapidly increased demands of integration and miniaturization continuously challenge energy densities of dielectric capacitors. New materials with high recoverable energy storage densities become highly desirable. Here, by structure evolution between fluorite $HfO_2$ and perovskite hafnate, we create an amorphous hafnium-based oxide that exhibits the energy density of ~155 J/ $cm^3$ with an efficiency of 87%, which is state-of-the-art in emergingly capacitive energy-storage materials. The amorphous structure is owing to oxygen instability in between the two energetically-favorable crystalline forms, in which not only the long-range periodicities of fluorite and perovskite are collapsed but also more than one symmetry, i.e., the monoclinic and orthorhombic, coexist in short range, giving rise to a strong structure disordering. As a result, the carrier avalanche is impeded and an ultrahigh breakdown strength up to 12 MV/cm is achieved, which, accompanying with a large permittivity, remarkably enhances the energy storage density. Our study provides a new and widely applicable platform for designing high-performance dielectric energy storage with the strategy exploring the boundary among different categories of materials.

Dielectric capacitors are fundamental for electric power systems, which store energy in the form of electrostatic field ($E$) against electric displacement ($D$, or polarization $P$), giving rise to fast charging/discharging rate and high power density far beyond other energy storage technologies, such as electrochemical capacitors and batteries[1–3]. However, the relatively low energy density, as a long-standing performance bottleneck, limits wide applications of dielectric energy-storage capacitors in advanced power systems.

In dielectrics, the energy storage density is regulated by essential material characters, the permittivity ($\varepsilon_r$) and breakdown field ($E_b$)[1,2].

The primary performance parameter—recoverable energy storage density ($U_{rec}$)—can be calculated by $\int_{P_r}^{P_m} EdP$, according to the $P$-$E$ hysteresis loop, which formulates the discharging upon $E$ from the remanent polarization ($P_r$) to the maximum polarization ($P_m$) before dielectric breakdown (Supplementary Fig. S1). Noting that the hysteresis area is the energy loss ($U_{loss}$) during a charging-discharging cycle. The $\eta$ is then obtained by $U_{rec}/(U_{rec} + U_{loss})$. For ideally linear dielectrics, the $U_{rec}$ can be written in a simplified form of $\frac{1}{2}\varepsilon_0\varepsilon_r E_b^2$ ($\varepsilon_0$: the vacuum permittivity)[1]. Therefore, a high-performance dielectric capacitor should hold both large $\varepsilon_r$ and high $E_b$, simultaneously. Moreover,

[1]College of Physics, Qingdao University, Qingdao 266071, China. [2]School of Physics, Shandong University, Ji'nan 250100, China. [3]College of Electronics and Information, Qingdao University, Qingdao 266071, China. [4]National Laboratory of Solid-State Microstructures, Department of Materials Science and Engineering, Jiangsu Key Laboratory of Artificial Functional Materials and Collaborative Innovation Center for Advanced Materials, Nanjing University, Nanjing 210093, China. [5]Institute of Micro-nano Photonics and Quantum Manipulation, School of Science, Nanjing University of Science and Technology, Nanjing 210094, China. [6]Department of Physics and Astronomy, Rutgers University, Piscataway, NJ 08854, USA. [7]These authors contributed equally: Yahui Yu, Qing Zhang, Zhiyu Xu. ✉e-mail: liuxiaohui@sdu.edu.cn; zwen@qdu.edu.cn

the increase of $E_b$ would be more efficient to improve the energy storage density due to the square dependence. However, $E_b$ is usually restricted by $\varepsilon_r$ in most dielectric materials, following a negative power law of $E_b \propto \varepsilon_r^{-\alpha}$ [1,4,5]. For example, perovskite oxides, such as $SrTiO_3$, $BaTiO_3$, and $Pb(Zr,Ti)O_3$, have large $\varepsilon_r$ of a few hundred but low $E_b$ of only 1.0–3.0 MV/cm in general[1,4,5]. For that have high breakdown strengths (>5.0 MV/cm), like $SiO_2$, polymers, and dielectric glasses, their low $\varepsilon_r$ limit energy densities[2,6,7]. How to overcome the negative correlation by increasing $E_b$ in the materials with large permittivity is key to enhance the energy storage performance.

Most recently, by introducing local disorders, such as grain boundaries, ionic defects, amorphous fractions, and interfacial layers, improved $E_b$ of 4.5, 5.92, 6.35, and 8.75 MV/cm have been achieved in $(Ba_{0.7}Ca_{0.3})TiO_3/Ba(Zr_{0.2}Ti_{0.8})O_3$ multilayers, ion-bombarded $Pb(Mg_{1/3}Nb_{2/3})O_3$-$PbTiO_3$, high-entropy $(Bi_{3.25}La_{0.75})(Ti_{3-x}Zr_xHf_xSn_x)O_{12}$, and nano-grained $BaTiO_3$, respectively, generating state-of-the-art energy storage densities with the perovskite oxides (Supplementary Table S1)[8–17]. For binary oxides, typified by high-κ $HfO_2$ family, which could be easily compatible with current complementary metal-oxide-semiconductor (CMOS) technology and are rapidly developed for the applications in the next-generation microelectronic power devices, the energy storage densities are relatively low with the $U_{rec}$ of only about tens of $J/cm^3$ [3,18–24].

Here, by employing a new structure-evolution strategy between fluorite $HfO_2$ and perovskite hafnate ($AHfO_3$, where $A$ is a divalent ion), we create an amorphous hafnium-based oxide that exhibits a breakdown strength as high as ~12 MV/cm. The $E_b$ is more than two times higher than the values reported to date in $HfO_2$-based films (Supplementary Table S1) and far exceeds the restriction of its permittivity (Supplementary Fig. S2). With the ultrahigh $E_b$, the $U_{rec}$ is remarkably improved to as high as ~155 $J/cm^3$ ($\eta = 87\%$), which is record-high in the high-κ binary oxides.

## Results

### Amorphization of hafnium-based oxides

As depicted in Fig. 1a, although they are classified into different categories of crystals, the $HfO_2$ and $AHfO_3$ share similar face-centered metal sublattices. The difference is the stoichiometric ratio and lattice sites of oxygen ions. In fluorite structure, the molar ratio of oxygen to metal is 2:1 and the eight oxygen ions occupy the interstitial sites of Hf tetrahedrons to support the Hf metal frame. For perovskite, the oxygen/metal molar ratio is reduced to 1.5:1 and the Hf/$A$ metal frame is stabilized by six oxygen ions that take the connection-line sites of two same metal ions, such as $Hf^{4+}$-$Hf^{4+}$ and $A^{2+}$-$A^{2+}$. Therefore, one can evolve the lattice from the fluorite to the perovskite by reducing oxygen stoichiometric ratio through substituting $Hf^{4+}$ with $A^{2+}$, in which the oxygen ions move from the interstitial to the connection-line sites. However, during the structure evolution, the oxygen ions would become very instable, which may distort the Hf/$A$ metal frames and eventually result in the collapse of long-range periodicity when the substitution concentration is proper.

As shown in Fig. 1b, alkaline-earth metals, such as $Ba^{2+}$, $Sr^{2+}$, or $Ca^{2+}$, are adopted to substitute $Hf^{4+}$ ions for driving the structure evolution. The substituted $HfO_2$ thin films are deposited on $SrTiO_3$ (STO) substrates buffered with epitaxial $(La_{0.67}Sr_{0.33})MnO_3$ (LSMO) as bottom electrodes by the means of pulsed laser deposition. X-ray diffraction (XRD) and scanning transmission electron microscopy (STEM) are employed to characterize the microstructures. Figure 1c demonstrates XRD patterns of the Ba-substituted $HfO_2$/LSMO/STO heterostructures (abbr. BHO$x$, where $x$ is the substitution concentration in percentage). The XRD for Sr- and Ca-substituted $HfO_2$ thin films are shown in Supplementary Fig. S3a and S4a, respectively.

For low concentration of $x \leq 4\%$, the Ba-Hf-O system is in fluorite structure, in which the BHO0 thin film (i.e., the undoped $HfO_2$) exhibits monoclinic ($m$) phase with a diffraction peak for the $(-111)_m$ reflection

observed at $2\theta = 28°$ while the BHO02 and BHO04 show the coexistence of $m$- and orthorhombic ($o$) phases because of the substitution-induced lattice strains, as evidenced by the presence of $(111)_o$ reflection at $2\theta = 30°$ [25,26]. In the substituted $HfO_2$, the tetragonal ($t$) phase may also coexist with the $o$-phase since the $(011)_t$ reflection is located at 30.05° and the $2\theta$ difference between $(111)_o$ and $(011)_t$ is too small to be distinguished[27]. Figure 1f demonstrates atomic-resolution high-angle annular dark-field (HAADF) images of the fluorite lattices in the $o$-phase, in which the fast Fourier transform of the BHO02 layer exhibits ordered diffraction spots (see Supplementary Fig. S5 for the area with two-phase coexistence). In addition, the element mappings of Hf, La, and Ba indicate a sharp interface between BHO and LSMO layers.

When $x \geq 6\%$, the structure evolution takes place. One can find in the XRD patterns that the Bragg reflections from the fluorite structure are quenched and no new diffraction peaks emerge in the BHO06–BHO15 thin films. Further characterization by the HAADF imaging indicates that there are no long-range periodicities observed in the representative BHO12 thin film (Fig. 1e) and the fast Fourier transform is a ring-shaped pattern, suggesting the formation of amorphous state, as expected in Fig. 1a. The presence of dispersion spots on the diffraction ring might be due to the short-range ordering within local regions of several atoms. More information about the BHO12/LSMO/STO heterostructure over a large scale is shown in Supplementary Fig. S6 for clarifying the uniformity of the amorphous structure. The amorphization is understood by performing first-principles calculation on the stability of oxygen ions, manifested by the formation energy of oxygen vacancy [$E^f(V_O)$] at the lattice sites. Note that the amorphous state is formed in a high-temperature crystallizing process of the Ba-Hf-O system (see "Methods"). The $HfO_2$ should have a high symmetry, like the $t$-phase in the space group of $P4_2/nmc$[26,28]. Figure 1g demonstrates the $E^f(V_O)$ at the 1st nearest-neighbor interstitial sites of Hf tetrahedrons in the tetragonal $HfO_2$ as a function of substitution concentration. As shown, $E^f(V_O)$ is as high as +8.25 eV in the undoped $HfO_2$, comparable with the previous reported values[29], but sharply lowered to −0.77 eV when one in 32 $Hf^{4+}$ ions are replaced by $Ba^{2+}$. With increasing concentration, $E^f(V_O)$ keeps low values around +0.2 to −1.0 eV. These suggest that oxygen ions are no longer favorable at the interstitial sites near the substituted $Ba^{2+}$ and oxygen vacancies ($V_O$s) are formed to maintain the electric neutrality. In addition, not only the 1st nearest-neighbor site but also the 2nd, 3rd, and 4th nearest-neighbor sites are all instable for oxygen ions even there are only 1/32 $Hf^{4+}$ ions are replaced (Fig. 1h), which may be due to the strong lattice distortion induced by the large difference in ionic radii between $Ba^{2+}$ (1.35 Å) and $Hf^{4+}$ (0.71 Å). Therefore, Ba substitution can dramatically affect the oxygen stability and efficiently reduce the oxygen stoichiometric ratio of $HfO_2$. At a proper substitution region, e.g., $4\% < x < 20\%$ in Fig. 1c, the number of oxygen ions is too less to support the fluorite Hf metal frame and the Ba-Hf-O system collapses into an amorphous state since the instability of oxygen ions destroy the long-range fluorite periodicity while the perovskite structure isn't formed yet in this oxygen/metal molar ratio.

The oxygen instability is also characterized by X-ray photoelectron spectroscopy (XPS). As shown in Supplementary Fig. S6, the $V_O$s increase with increasing $x$ from 0% to 12% since the reduction of oxygen stoichiometric ratio from the fluorite to the amorphous structure. However, with further increasing the Ba concentration, $V_O$s are decreased in the BHO20 and become negligible in the BHO50 (i.e., the $BaHfO_3$). These suggest that, as the oxygen/metal molar ratio is further reduced, the Ba/Hf metal frame evolves to the perovskite type that requires less oxygen ions to be stabilized. During the structure evolution, both the Ba and Hf keep oxidation state, as shown in XPS spectra of the amorphous BHO12 (Supplementary Fig. S8). The structure evolution is also consistent with the XRD patterns. When the Ba concentration is increased to $x \geq 20\%$, two diffraction peaks along with the $(00l)$ reflections of STO emerge and become stronger from BHO20

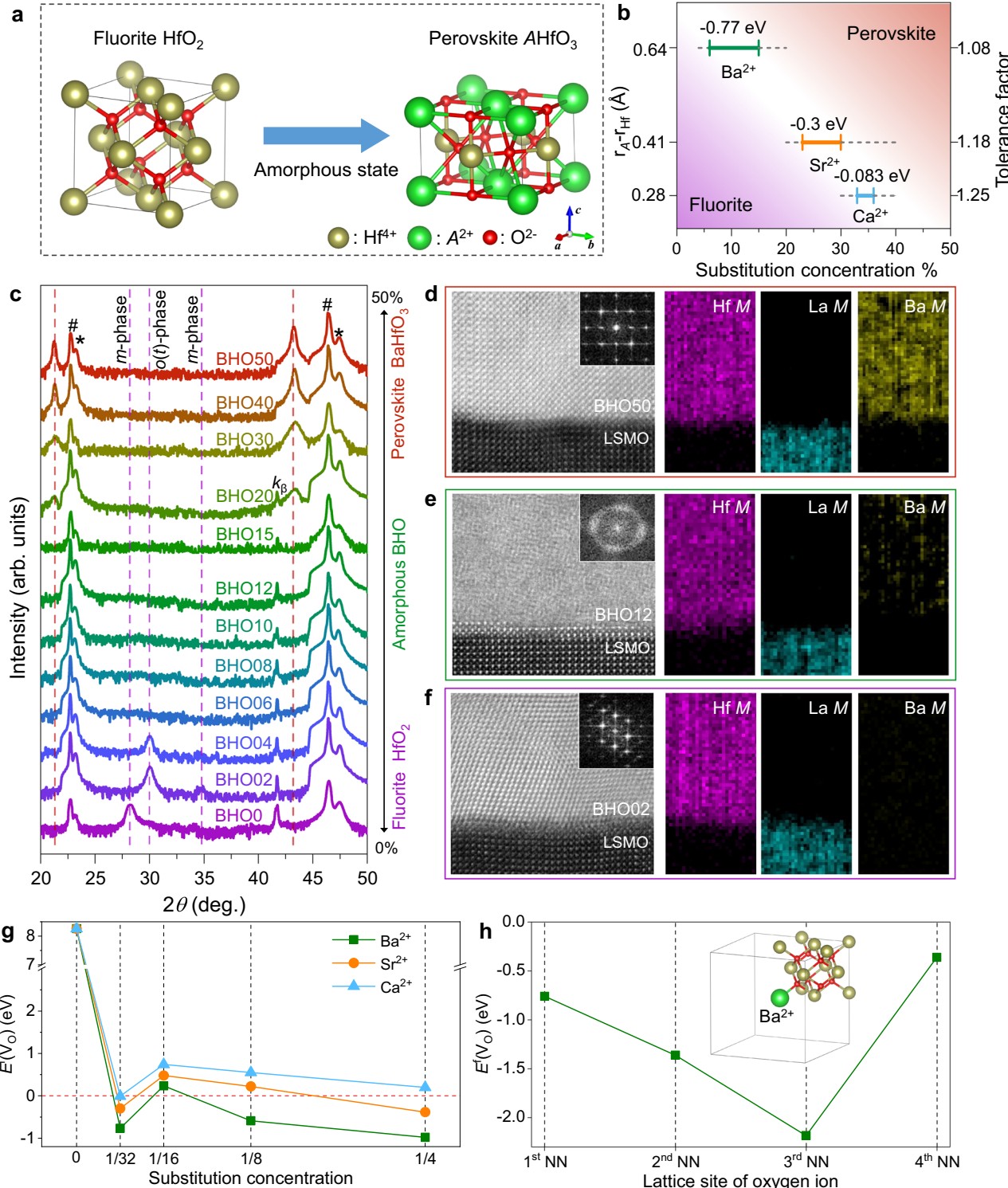

**Fig. 1 | Amorphization of hafnium-based oxides. a** Schematic drawing for the structure evolution from fluorite $HfO_2$ to perovskite $AHfO_3$, where the $HfO_2$ is drawn in normal coordinates of <100> (*a* axis), <010> (*b* axis), and <001> (*c* axis) while the $AHfO_3$ is drawn in the coordinates of <110> (*a* axis), <1$\bar{1}$0> (*b* axis), and <001> (*c* axis). **b** Amorphous regions of the Ba-Hf-O, Sr-Hf-O, and Ca-Hf-O systems, respectively, as functions of the difference in ionic radii between $A^{2+}$ and $Hf^{4+}$ ($r_A$-$r_{Hf}$) and the tolerance factor of $AHfO_3$. **c** XRD patterns of Ba-substituted $HfO_2$ (BHO$x$) thin films with increasing concentration from 0 to 50%. The # and * symbols denote Bragg reflections from STO substrate and epitaxial LSMO electrode, respectively. The purple and red dashed lines indicate Bragg reflections from fluorite (*m*- and

*o*(*t*)-phases) and perovskite structures, respectively. The peak at ~42° is $k_\beta$ of STO substrate. STEM characterizations of the BHO50 (**d**), BHO12 (**e**), and BHO02 (**f**) heterostructures, where the left panels are high-resolution HAADF images with fast Fourier transform patterns shown in the insets and the right panels are element distributions of Hf, La, and Ba mapped by electron energy loss spectroscopy, respectively. **g** Formation energy of oxygen vacancy [$E^f$($V_O$)] at the first nearest-neighbor (NN) site as a function of the substitution concentration in *t*-phase $HfO_2$. **h** $E^f$($V_O$) at different nearest-neighbor sites for the Ba concentration of 1/32. The inset depicts the lattice structure.

to BHO50. The epitaxy of perovskite BHO50 on LSMO/STO is further demonstrated by the HAADF imaging in Fig. 1d.

In Fig. 1g, we also show the $E^f(V_O)$ of Sr- and Ca-substituted HfO$_2$, which are −0.3 and −0.083 eV at the concentration of 1/32, respectively, higher than that of the Ba-Hf-O system. The decrease of oxygen instability may be due to the smaller difference in ionic radii between $A^{2+}$ (Sr$^{2+}$: 1.12 Å; Ca$^{2+}$: 0.99 Å) and Hf$^{4+}$. It also coincides with the structure evolution shown in XRD patterns, in which the Sr-Hf-O and Ca-Hf-O systems need higher substitution concentrations to induce the amorphous structures in $23\% \leq x \leq 30\%$ and $33\% \leq x \leq 36\%$, respectively (Supplementary Figs. S3a and S4a), compared with the Ba-Hf-O. The amorphization behaviors are summarized in Fig. 1b. As shown, in the $A$-Hf-O system both width and location of amorphous region could be controlled by the substituted ion through the difference in ionic radius ($r_A - r_{Hf}$) and the tolerance factor of the formed perovskite $A$HfO$_3$, calculated by $\frac{\sqrt{2}(r_{Hf} + r_O)}{r_A + r_O}$.

## Short-range ordering of the amorphous BHO

Short-range ordering of the designed amorphous structure is characterized by extended X-ray absorption fine-structure spectroscopy (EXAFS). Figure 2 demonstrates the Fourier transformed EXAFS data (|$\chi(R)$|) of Hf $L_{III}$ edge for the representative BHO12 film, where $R$ denotes the radial distance. In the EXAFS spectrum, the oscillations originate from the X-ray scattering between the photon-absorbing Hf atoms and their neighboring O (or Hf) atoms. By fitting the experimental data to possible lattice models, the coordination structure can be extracted. Previous density-functional-theory and EXAFS studies have shown that amorphous HfO$_2$ films are always monoclinic in local structure with the best fit to the $P2_1/c$ symmetry[30-33]. For comparison, we fabricate an amorphous 12% Ba-substituted HfO$_2$ at room temperature (BHO12-RT) and find that the amorphous film also exhibits the short-range $P2_1/c$ symmetry, as evidenced by the fitting within $R = 1.0–4.0$ Å in the inset, which is in agreement with the reported results[30-33]. However, the best fit to the energetically-favorable phase of bulk HfO$_2$ suggests that the Ba substitution doesn't yield pronounced structure distortion on the BHO12-RT film, which may be explained by more stable oxygen ions in the conventionally amorphous structure (Supplementary Fig. S9).

Following the scattering paths used in BHO12-RT, the $P2_1/c$ symmetry cannot give a good fit to the BHO12 film mainly because of the two distinguished oscillations in $2.2$ Å $< R < 3.5$ Å. A better fit can be found in orthorhombic $Pca2_1$ symmetry and the best is achieved by combining the $Pca2_1$ and $P2_1/c$, which is reasonable since, before the collapse of long-range fluorite periodicity, the Ba-Hf-O system has experienced an orthorhombic distortion. Similar two-phase coexistence has also been observed in the EXAFS spectrum of Hf$_{0.46}$Zr$_{0.54}$O$_2$ films but they are crystalline[33]. Therefore, the observation of pronounced $Pca2_1$ symmetry in Fig. 2 suggests that the Ba substitution-induced lattice distortion can be preserved in the short-range structure of BHO12, which isn't fully-relaxed like the unannealed BHO12-RT counterpart. Based on the fitting, the coordination information can be extracted (see Supplementary Text 1 for details). The BHO12 exhibits a Hf-O bond length of 2.07–2.09 Å, which is shorter than that of the BHO12-RT and the previously reported amorphous HfO$_2$, as well as the average Hf-O interatomic distance of crystalline HfO$_2$ (~2.14 Å)[28,31,32], indicating a higher density. More importantly, due to the coexistence of $Pca2_1$ and $P2_1/c$ symmetries, the BHO12 shows a strong short-range disordering, in which the disorder (Debye-Waller) factor is as large as ~0.011, higher than both the unannealed one and the amorphous HfO$_2$ in literature[30-32]. These structure characters are beneficial to achieve high breakdown strengths.

## Dielectric energy storage properties with structure evolution

Pt is adopted as top electrodes for fabricating dielectric capacitors. Figure 3a shows $P$-$E$ hysteresis loops of the Pt/BHO/LSMO capacitors and the corresponding Weibull distributions of breakdown strengths

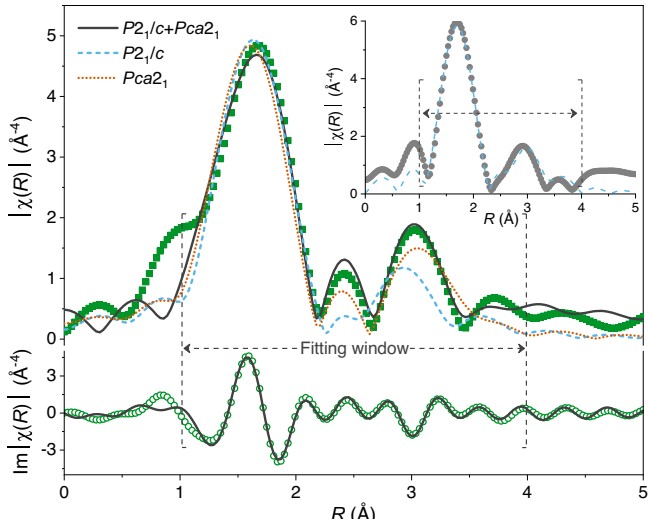

**Fig. 2 | Short-range structure of the amorphous BHO film.** The Fourier transformed EXAFS data (|$\chi(R)$|) of Hf $L_{III}$ edge for BHO12 film, in which the imaginary part of |$\chi(R)$| is also shown for clarity. The inset is the EXAFS spectrum of BHO12-RT film for comparison. The bule dashed, orange dotted, and black solid lines are fits to $P2_1/c$, $Pca2_1$, and $P2_1/c+Pca2_1$ symmetries, respectively. The fitting window is $R = 1.0–4.0$ Å, as indicated by the dashed arrows and square brackets.

are plotted in Fig. 3b. Without Ba substitution, the BHO0 is a linear dielectric with the statistical $E_b$ of ~4.2 MV/cm (Fig. 3c), in agreement with the values reported previously in similar HfO$_2$ thin films[3,19]. The calculated $U_{rec}$ is only ~22.4 J/cm$^3$ (Fig. 3e) due to the low $E_b$ and $P_m$. With Ba substitution, the polar $o$-phase is induced and a typical ferroelectric hysteresis loop is observed in the BHO02 capacitor. $U_{rec}$ is increased to ~32 J/cm$^3$ but the strong hysteresis feature results in a large $U_{loss}$ and thus a low $\eta$ of ~37%. Further increasing Ba substitution to $x = 4\%$, although the BHO04 is still dominated by the $o$-phase (Fig. 1c), the ferroelectric character becomes weak, which may be ascribed to the formation of amorphous fractions since, according to the first-principles calculation, a low Ba concentration of 1/32 (~3.1%) can result in remarkable instability of the neighboring oxygen ions. Both the $U_{rec}$ and $\eta$ are somewhat improved due to the reduced hysteresis in the $P$-$E$ loop.

Above $x = 6\%$, the Ba-Hf-O system evolves into amorphous state. The hysteresis behaviors become very weak and hence the $\eta$ is increased to above 85% (Fig. 3e). The $U_{rec}$ is also substantially increased. It increases to ~100 J/cm$^3$ in the BHO08 and reaches a maximum value of ~155 J/cm$^3$ in the BHO12. In the BHO15, the $U_{rec}$ is relatively decreased but still maintains a large value above 120 J/cm$^3$. The giant energy densities are obviously owing to the dramatically improved breakdown strengths in the amorphous capacitors (Fig. 3c). Taking the BHO12 as an example, its $E_b$ can be as high as ~12 MV/cm, about three times of that of the crystalline BHO0, yielding a large $P_m$ of ~30 μC/cm$^2$. In addition, the amorphous BHO also exhibits large Weibull modulus $\beta$, indicating good reproducibility over different samples. However, when $x$ further increases to above 20%, the perovskite BaHfO$_3$ is crystallized and $E_b$ is decreased to less than 7.0 MV/cm, resulting in low $U_{rec}$ of 50–65 J/cm$^3$ in the BHO20–BHO50 capacitors.

Overall, Fig. 3 indicates the critical role of breakdown strength for enhancing energy storage density. In dielectric capacitors, the breakdown usually takes place within a short period of time (<1.0 ms) and results from the electronic and/or the avalanche mechanisms[1,2]. Considering that the BHO thin films have similar bandgaps of ~5.0 eV (Supplementary Fig. S10)[34,35], the electronic breakdown that is due to the activation of electrons from the valence band to the conduction band by $E$ may not be a major origin responsible for the remarkable

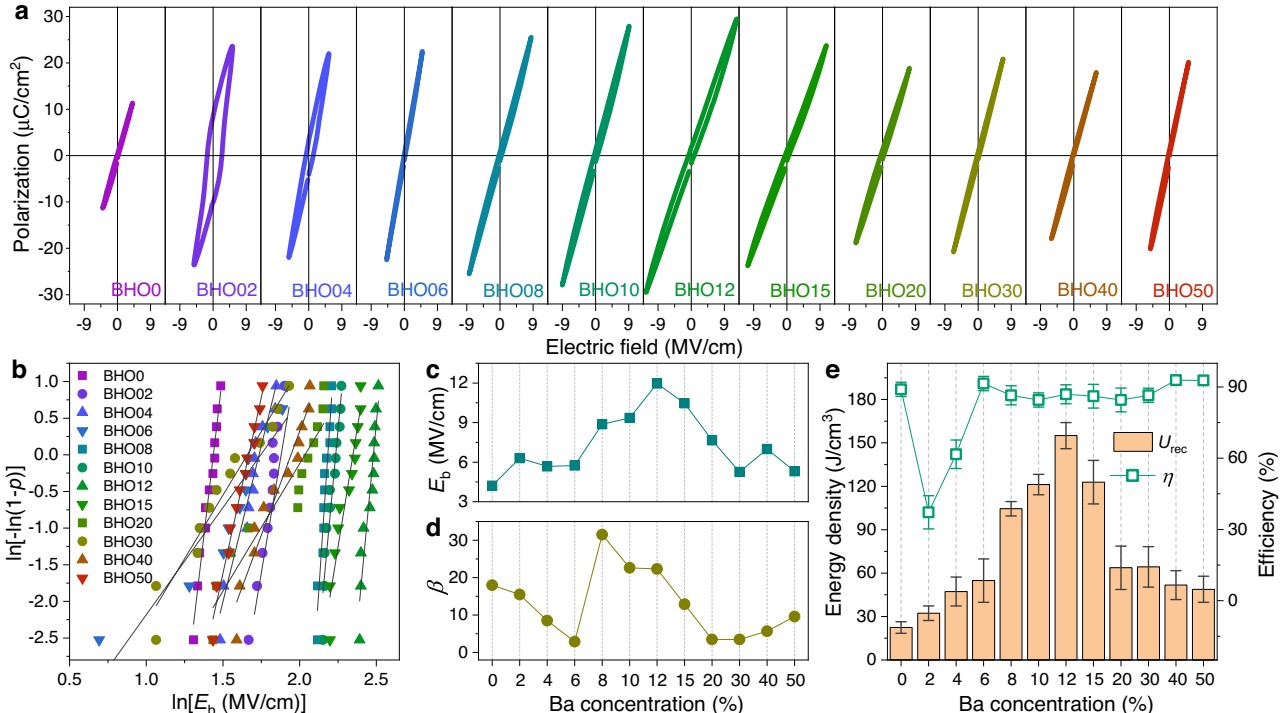

**Fig. 3 | Dielectric energy storage of BHO thin-film capacitors. a** *P-E* hysteresis loops of Pt/BHO/LSMO capacitors measured at 10 kHz. **b** Two-parameter Weibull distribution analysis of breakdown strengths over 12 capacitors for each Ba concentration. **c** Statistical $E_b$ and (**d**) Weibull modulus $\beta$ extracted from (**b**) plotted as a function of Ba concentration. Here, $\beta$ is the slope of $\ln[-\ln(1-p)]$ vs. $\ln E_b$, where $p = i/(n+1)$ ($n$ is the total number of samples and $i$ is the $i$th sample). **e** Energy storage density ($U_{rec}$) and efficiency ($\eta$) of the BHO capacitors calculated from *P-E* loops. The error bars are averaged over 12 capacitors for each Ba concentration.

different in breakdown strength among the BHO capacitors[36]. The improved $E_b$ is thus ascribed to the suppression of avalanche effect in the amorphous BHO capacitors. First, the amorphous BHO is formed in the structure evolution by oxygen instability, which exhibits a strong disordering not only due to the collapse of fluorite and perovskite periodicities in long range but also the coexistence of $Pca2_1$ and $P2_1/c$ symmetries in short range. Second, the high-temperature annealing but non-crystallization gives the BHO a higher density than the reported crystalline/amorphous $HfO_2$ and the unannealed counterpart (e.g., the BHO12-RT, showing an $E_b$ of ~3.64 MV/cm and a low $U_{rec}$ of ~10.4 J/cm³, Supplementary Fig. S11). In this highly-disordered and dense matrix, the carrier transport is dramatically scattered from one lattice to the other, which suppresses the ionizing collision effect with atoms and hence impedes the carrier avalanche for dielectric breakdown. One can thus find that the amorphous BHO12 capacitor exhibits a negligible dependence of $E_b$ upon film thickness whereas the $E_b$ of crystalline BHO0, BHO02, and BHO50 capacitors decrease with increasing thickness (Supplementary Fig. S12a). The thickness-dependent $E_b$ is further analyzed by the 40-generation-electron theory[1], as shown in Supplementary Fig. S12b. In parallel with the improved breakdown behavior, the amorphous BHO also exhibits a superior insulating character with a leakage current density <$1.0 \times 10^{-6}$ A/cm² (Supplementary Fig. S13). On the other hand, in the amorphous structure, the bonding of Hf-O could be well maintained for contributing the dielectric polarizability (as seen in Supplementary Fig. S8, the Hf has a valence of four). A large $\varepsilon_r > 18$ is obtained in the BHO12 (also in the amorphous state of Sr-Hf-O and Ca-Hf-O systems), which is even higher than that of the crystalline BHO0 at high frequency (Supplementary Fig. S14). Therefore, the ultrahigh breakdown strength that is achieved without the trade-off of permittivity gives rise to the remarkably improved energy density in the amorphous hafnium-based oxide.

**Energy storage performance of the BHO dielectric capacitors**

Energy storage performances of the amorphous BHO12 are further characterized by comparing with crystalline BHO0, BHO02, and BHO50 capacitors. Figure 4a plots the $U_{rec}$ and $\eta$ as a function of $E$. The BHO12 capacitor exhibits a parabolic-like increase of $U_{rec}$ to 155 J/cm³ with small variation in $\eta$ up to 12 MV/cm. However, in the BHO0, BHO02, and BHO50 capacitors, the dielectric breakdown occurs before 6.0 MV/cm, impeding the increase of $U_{rec}$. Corresponding *P-E* loops are shown in Supplementary Fig. S15 for clarity. Owing to the improved breakdown strength, the BHO12 capacitor exhibits much higher energy storage densities in reliability measurements. As shown in Fig. 4b, at $E = 0.7E_b$, the $U_{rec}$ of BHO12 can be 78–86 J/cm³, which is about 3 times of that of the BHO0, BHO02, and BHO50, over the frequency range of 1–20 kHz. Figure 4c shows the charging/discharging endurance at the cycling $E = 0.6E_b$, in which the BHO12 exhibits optimized energy storage properties up to $2.0 \times 10^6$ cycles with a large $U_{rec}$ of ~56 J/cm³ and a high $\eta$ of ~90% at 7.2 MV/cm. In addition, the amorphous BHO12 holds a similar temperature stability with that of the crystalline BHO0 and BHO50 but exhibits a more than two times higher $U_{rec}$ of ~80 J/cm³ ($\eta = 84\%$) at 400 K (Fig. 4d). In Fig. 4a–d, one may also notice the obvious changes in the $\eta$ of BHO02 capacitor, which is due to its ferroelectric behaviors, as discussed in Supplementary Fig. S15. Figure 4e demonstrates the representative $U_{rec}$ and $E_b$ reported to date in emergingly capacitive energy-storage materials, which includes the state-of-the-art results for each oxide system (see Supplementary Table S1 for more details). As shown in the solid symbols, we achieve the highest energy density in the community of high-κ binary oxides. More importantly, owing to the ultrahigh $E_b$, the $U_{rec}$ of amorphous BHO12 is also higher than that of most perovskite oxides (the hollow symbols) even though they have the permittivity up to thousands, about two orders larger than the $HfO_2$-based oxides.

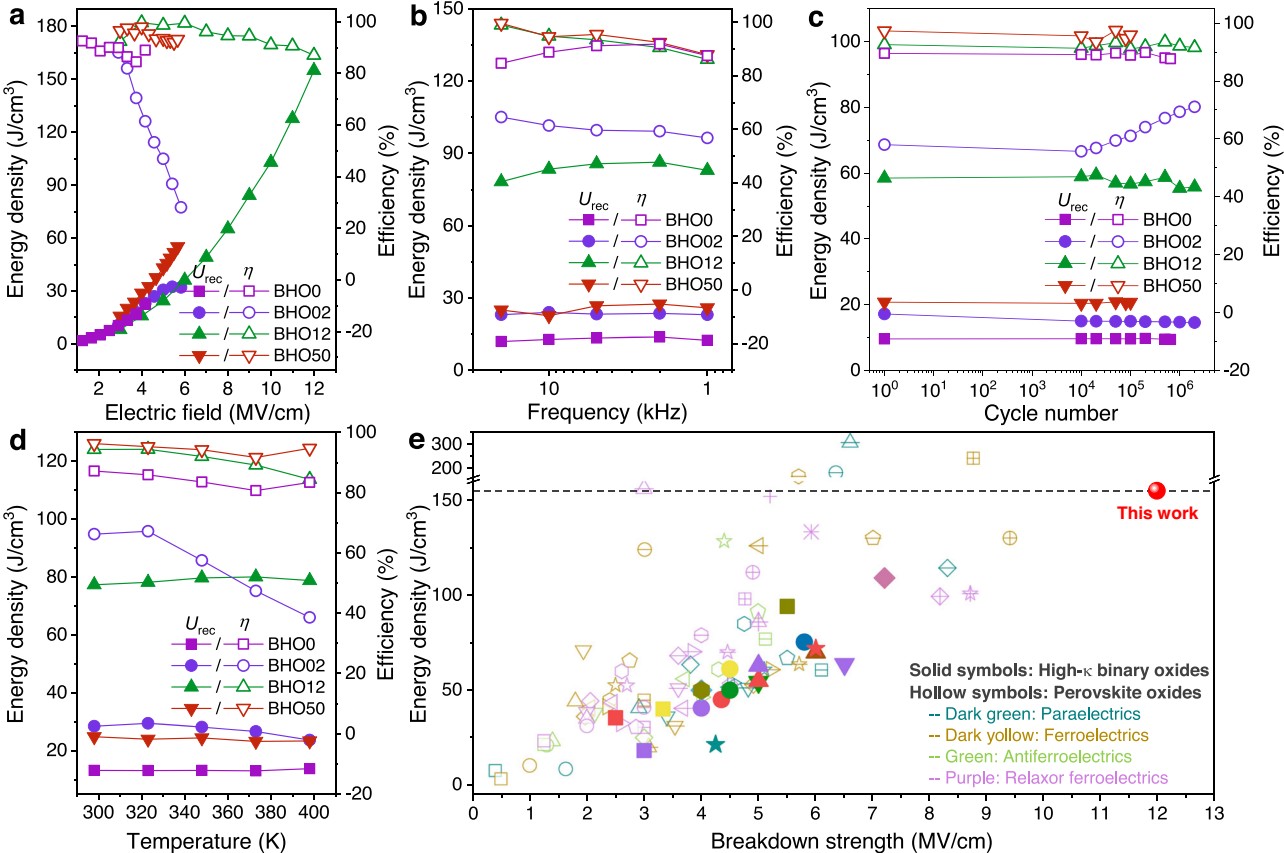

**Fig. 4 | Energy storage performance of the BHO dielectric capacitors.** Energy storage density ($U_{rec}$) and efficiency ($\eta$) of the BHO0, BHO02, BHO12, and BHO50 capacitors as functions of **a** electric field, **b** frequency (measured at 0.7 $E_b$), **c** charging-discharging cycles (measured at 0.6 $E_b$), and **d** temperature (measured at 0.7 $E_b$). **e** Comparison of the $U_{rec}$ and $E_b$ of our amorphous hafnium-based oxide (BHO12) with representative dielectrics covering high-κ binary oxides and perovskite oxides, in which the data are plotted from Supplementary Table S1.

## Discussion

We have proposed a structure-evolution strategy to create amorphous hafnium-based oxides by bridging fluorite HfO$_2$ and perovskite hafnate. In the amorphous structures, the oxygen instability results in strong disordering in both short and long ranges and hence yields ultrahigh breakdown strengths, which, accompanying with the large permittivities, give rise to giant energy storage densities. In addition to the $E_b$ = 12 MV/cm in the Ba-Hf-O system, the amorphous Sr-Hf-O and Ca-Hf-O also show very high $E_b$ of ~10 and ~8.3 MV/cm, yielding large $U_{rec}$ of ~117 and ~72 J/cm$^3$, respectively (Supplementary Figs. S3b and S4b). More interestingly, the amorphization behaviors are found to depend upon intrinsic material parameters of the HfO$_2$ and the alkaline-earth perovskites, such as ionic radii and tolerance factor shown in Fig. 1b. These findings suggest highly controllable and widely applicable of the amorphous state with the variety of fluorite and perovskite structures, which provides a new platform for designing high-performance dielectric energy storage.

Besides, from a practical point of view, the hafnium-based oxide, which is high-κ but shows ultrahigh $E_b$ comparable to the SiO$_2$ (Supplementary Fig. S2), would be promising in a broad spectrum. Especially, its amorphous character that is formed by composition engineering and can be stable at a high-temperature process window has great potential to be compatible with CMOS techniques for developing advanced electronic devices that require high breakdown strengths[37–39]. More generally, the structure-design approach proposed in this work may open a new perspective for exploring new functionalities in the boundary among different categories of materials.

## Methods

### Device preparation

The $A$-Hf-O thin films and LSMO electrodes were grown on (001) single-crystalline STO substrates by pulsed laser deposition using a KrF excimer laser (Coherent COMPexPro 201). The LSMO thin films were deposited at a laser energy density of ~3.0 J/cm$^2$ with a repetition rate of 2 Hz, keeping the substrate at 973 K and the oxygen pressure at 0.2 mbar. The $A$-Hf-O thin films were deposited with 2.6 J/cm$^2$ laser energy density at 4 Hz repetition, keeping the substrate temperature at 873 K and the O$_2$ pressure at 0.1 mbar. After the deposition, the $A$-Hf-O heterostructures were annealed at 973 K for 1 hour in flowing O$_2$. Pt top electrodes of ~30 μm in diameter and ~50 nm in thickness were deposited on the surface of $A$-Hf-O heterostructures by sputtering with a shadow mask to form the thin-film capacitors.

### First-principles calculation

Density-functional theory (DFT) calculations are performed using Quantum ESPRESSO. The exchange and correlation effects are treated within the generalized gradient approximation (GGA) of Perdew−Burke−Ernzerhof (PBE). The Brillouin zone is sampled with 6 × 6 × 6 Monkhorst-Pack k-point meshes for the conventional unit cell of HfO$_2$ which is reduced reciprocally for larger supercells. The electronic wave functions are expanded in a plane-wave basis set limited by a cut-off energy of 900 eV. The atomic positions and lattice parameters are optimized until the force on each atom is converges to <1.0 meV/Å in all supercells.

As shown in the table below, the lattice parameters of the three phases of HfO$_2$ are calculated which agree well with the experimental results (see Table 1 for details)[40–43]. In order to study the effects of

**Table 1 | Lattice constant used in the DFT calculations**

|  | Calculation | Experiment[40–43] |
|---|---|---|
| Monoclinic |  |  |
| a | 5.11 Å | 5.12 Å |
| b | 5.15 Å | 5.17 Å |
| c | 5.29 Å | 5.30 Å |
| β | 99.65° | 99.20° |
| Orthorhombic |  |  |
| a | 5.24 Å | 5.23 Å |
| b | 5.01 Å | 5.00 Å |
| c | 5.05 Å | 5.05 Å |
| Tetragonal |  |  |
| a | 5.04 Å | 5.15 Å |
| c | 5.20 Å | 5.29 Å |

alkaline-earth metal $Ba^{2+}$ (or $Sr^{2+}$, $Ca^{2+}$) doping on the structural stability of $HfO_2$, we construct several supercells: $1 \times 1 \times 1$, $\sqrt{2} \times \sqrt{2} \times 1$, $\sqrt{2} \times \sqrt{2} \times 2$, and $2 \times 2 \times 2$ unit cells which include 4, 8, 16, and 32 $Hf^{4+}$ ions, respectively. With one $Hf^{4+}$ substituted by one alkaline-earth metal ion, we could simulate different doping concentrations of 1/4, 1/8, 1/16, and 1/32.

The oxygen vacancy formation energy is defined by $E^f(V_O) = E_{defect} - E_{pure} + \mu_o$. In our calculation, $E_{defect}$ is the total energy of a supercell containing a $Ba^{2+}$ (or $Sr^{2+}$, $Ca^{2+}$) ion and an oxygen vacancy; $E_{pure}$ is the total energy for the equivalent supercell substituted with a $Ba^{2+}$ (or $Sr^{2+}$, $Ca^{2+}$) ion, and $\mu_o$ is the chemical potential of oxygen atom ($\mu_o = \mu_{o_2}/2$). The calculation is under the oxygen rich condition by considering that the annealing process in experimental is under the flowing $O_2$.

### Characterizations

XRD was performed on a Rigaku SmartLab diffractometer. The cross-sectional TEM specimens were prepared by focused ion beam (FIB, FEI Versa workstation) with a Ga ion source. The HAADF-STEM images were carried out at 200 kV by a JEOL ARM200CF microscope equipped with a cold field emission electron gun, an ASCOR probe corrector, and a Gatan Quantum ER spectrometer. The Hf $L_{III}$-edge X-ray adsorptions were measured on 150-nm-thick BHO films in fluorescence yield mode at room temperature at the BL14W1 beamline in the Shanghai Synchrotron Radiation Facility (SSRF). The EXAFS spectra were analyzed using FEFF6 code by Athena and Artemis packages[44]. XPS was performed on a Thermo ESCALAB 250 Xi, and the binding energy was calibrated by setting the C 1s at 284.6 eV. P-E hysteresis loops were measured by a Radiant Premier II ferroelectric tester. Capacitances were recorded using an Agilent 4294 A impedance analyzer. The testing pulses were applied to the Pt electrodes and the LSMO were always grounded.

### Data availability

The data that support the findings of this work are available within the article and its Supplementary Information file. Source data are provided with this paper.

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

## Acknowledgements

This work was jointly sponsored by National Natural Science Foundation of China (51872148 and 11974211), Natural Science Foundation of Shandong Province (ZR2020JQ03), the Taishan Scholar Program of Shandong Province (tsqn201812045), and Qilu Young Scholar Program of Shandong University.

## Author contributions

Z.W. and X.L. conceived this work and designed the experiments and calculations. Q.Z. carried out the first-principles calculations. Y.Y., Z.Y.X., W.Z., and C.Z. deposited the heterostructures, fabricated the capacitors, and measured the energy storage properties. Y.Y., Z.N.X., and J.X. collected the XRD data. C.D. measured the capacitances. L.Z. and Y.C. performed the XPS measurements. Z.W. performed the STEM and EXAFS analyses. Z.W., X.L., Y.Q., S.L., A.L., D.W., and K.M.R. analyzed the experimental data and the first-principles calculation. Z.W., X.L., Y.Y., Q.Z., and K.M.R. wrote the manuscript. All authors discussed the data and contributed to the manuscript.

## Competing interests

The authors declare no competing interests.
