## [Peer Review File · Nature Communications]

Structure-evolution-designed amorphous oxides for dielectric energy storageREVIEWER COMMENTS

Reviewer #1 (Remarks to the Author):

The manuscript by Yu et al. reported a giant energy storage density of $\sim 155 \text{ J/cm}^3$ in hafnium-based oxides by an “amorphous”-design method. The “amorphous” hafnium oxide is created by a controllable structure evolution from the fluorite HfO_2 to the perovskite hafnate BaHfO_3 , SrHfO_3 , or CaHfO_3 . The instability of oxygen ions could be controlled by doping Ba, Sr or Ca, which destroys long-range periodicities and leads to a complicated structure in short-range. The strong disordering results in significantly improved breakdown strength up to 12 MV/cm , which is impressive in this field for dielectric energy storage. Especially, the structural-evolution strategy proposed in this work is novel, which provides a new perspective for material design and is also valuable to the research in related electronic fields. The results are solid and the manuscript is well organized and written. Therefore, I would like to recommend it for publication in Nature Communications after the following issues being addressed by the authors properly.

- 1) The authors claimed that the energy density achieved in the “amorphous” hafnium oxide is record-high in high-k materials. Probably, this statement is inappropriate. In general, SrTiO_3 and BaTiO_3 can also be classified into high-k materials, where higher energy densities have been reported, as listed in Table S1. In Fig. 4e, the binary oxides, such as HfO_2 and ZrO_2 systems, may be described as the high-k materials that are easily compatible with the current CMOS process. It is better emphasize this point clearly.
- 2) In the BHO02 thin film, the authors have described the coexistence of m- and o- phases. However, in the HAADF image, I only found the results of o-phase. The authors should provide the STEM results for the m-phase of the BHO02.
- 3) Also, in the structure characterizations in Fig. 1, I found that the amorphous regions of Ba-Hf-O are strongly non-stoichiometric. Noting that these samples have also been treated by the same annealing temperature for the crystalline HfO_2 and BaHfO_3 . Why are there no secondary or impurity phases formed in the structural evolution range?
- 4) In the amorphous structure, the metal-oxygen bonding may differ from that of the crystalline form. It is thus necessary to characterize the valence state of Hf and Ba ions.
- 5) In Fig. 3a, compared with the BHO02, I found that the hysteresis of BHO04 capacitor becomes weaker but this composition exhibits a more pronounced ferroelectric o-phase in the XRD patterns. Why?
- 6) About the discussion of dielectric breakdown properties, it is better to measure leakage currents of the BHO capacitors as well, and include it in the paper.
- 7) It is also related to the discussion of breakdown. In this paragraph, the wording “Considering that the BHO thin films....., the electronic breakdown.....can be excluded” is inappropriate. In the BHO capacitors, the electronic breakdown cannot be excluded since it is an intrinsic character of dielectrics. In fact, what could be excluded in this case is that the electronic breakdown may be not a dominant factor for the difference in breakdown strength between the amorphous and crystalline BHO capacitors.
- 8) Regarding the analysis of breakdown field with thickness, the negative power law used in Fig. S10 is a general description for breakdown behaviors of dielectrics. I suggest the authors employing specific formula of the avalanche mechanism to analyze the thickness dependent breakdown strength, such as the “40-generation-electron theory” mentioned in Ref. 1, which probably be better.
- 9) For the calculation of formation of oxygen vacancy, there are oxygen rich and poor conditions. It seems like that the oxygen poor condition are considered. How does that calculation match the experimental environment?

Reviewer #2 (Remarks to the Author):

The manuscript by Yu et al. reported amorphous hafnium-based oxides for dielectric energy storage. They created amorphous hafnium-based oxides by bridging fluorite HfO₂ and perovskite hafnate. The structure evolution with different compositions was investigated by XRD and TEM. The formation mechanism of amorphization was tried to understand by DFT calculations. A breakdown field strength (E_b) of ~12 MV/cm and a recoverable energy storage density (U_{rec}) of ~155 J/cm³ were successfully obtained in this material system. This work proposed a structure-evolution strategy to overcome the negative correlation between E_b and permittivity. It may guide the research of energy storage materials and expand the application field of hafnia-based oxides, while some scientific problems should be solved.

1. In Table S1, the thickness in different samples is varied. What's the influence of the thickness to the structure evolution and energy storage performance?
2. In Figure 1c, the diffraction peak at 30 degree was labeled as o-phase. But the diffraction peak of tetragonal (t-) phase is also around 30 degree. Why the authors exclude the possibility of t-phase in the film. Why the peak of BHO0 at 30 degrees is not significant, but the peak at 28 degree is more significant than that in BHO02 and BHO04. What does the peak at 42 degrees stands for, it seems that the intensity gradually reduced from bottom (Fluorite HfO₂) to top (Perovskite BaHfO₃)?
3. The authors said that the amorphous BHO was formed in a high-temperature (973 K) crystallizing process and the HfO₂ should have a high symmetry, like the cubic (c-). Then, they adopted c-phase in the DFT calculation (Fig. 1g, h). Based on the phase diagram of HfO₂, c-phase is an ultra-high temperature phase. At 973K, the HfO₂ should have a m-symmetry without dopants and pressure, or have a t-symmetry with dopants or pressure. The authors need to reevaluate the calculation result.
4. In Figure 1e, small crystalline clusters in HAADF images and texture spot in FFT patterns can be found, did this suggest that the film is not in the amorphous state. Since the TEM image reflects a local area, more TEM evidences should be provided. Besides, the elemental distribution for Hf and Ba in Figure 1e seems also not very uniform, especially for Ba element as shown in Figure S5, is this the reason for the change in performance?
5. In Figure 1g, h and Figure S7, why did the authors only choose the c-phase, m-phase and o-phase in the DFT calculations. The t-phase should also be considered.
6. The two fitting curves for the EXAFS data do not match the experimental results very well, especially in the upper curves.
7. In the last paragraph of "Introduction", the authors took a lot of words to describe the crystal structure of HfO₂ and AHfO₃ and the strategy of the amorphous structure designing in Fig. 1a. These contents seem inappropriate to be described in "Introduction", but in the first part of "Results". Besides, there were many detailed and conclusive sentences about the physical mechanisms. However, these conclusions can only be reached after describing the experimental results and thus should be moved to the "Conclusions" or "Discussions"

Response to Reviewer #1

The manuscript by Yu et al. reported a giant energy storage density of ~ 155 J/cm³ in hafnium-based oxides by an “amorphous”-design method. The “amorphous” hafnium oxide is created by a controllable structure evolution from the fluorite HfO₂ to the perovskite hafnate BaHfO₃, SrHfO₃, or CaHfO₃. The instability of oxygen ions could be controlled by doping Ba, Sr or Ca, which destroys long-range periodicities and leads to a complicated structure in short-range. The strong disordering results in significantly improved breakdown strength up to 12 MV/cm, which is impressive in this field for dielectric energy storage. Especially, the structural-evolution strategy proposed in this work is novel, which provides a new perspective for material design and is also valuable to the research in related electronic fields. The results are solid and the manuscript is well organized and written. Therefore, I would like to recommend it for publication in Nature Communications after the following issues being addressed by the authors properly.

We appreciate your time and effort for reviewing our manuscript, and cordially thank you for your comments that the structure strategy proposed in this work is novel and provides a new perspective for material design in electronic fields. All your comments have been addressed point-by-point as shown below and the manuscript has been revised correspondingly. The revisions are highlighted in blue in the revised manuscript.

1) The authors claimed that the energy density achieved in the “amorphous” hafnium oxide is record-high in high-k materials. Probably, this statement is inappropriate. In general, SrTiO₃ and BaTiO₃ can also be classified into high-k materials, where higher energy densities have been reported, as listed in Table S1. In Fig. 4e, the binary oxides, such as HfO₂ and ZrO₂ systems, may be described as the high-k materials that are easily compatible with the current CMOS process. It is better to emphasize this point clearly.

In this work, we refer the high- κ materials to the HfO₂-based (including ZrO₂-

based) binary oxides that could be compatible with the current CMOS process and are also utilized in commercial devices. We therefore classified SrTiO₃ and BaTiO₃ into the paraelectric and ferroelectric perovskites. However, in a broad sense, the high-κ terminology can also mean the material having permittivity higher than that of SiO₂. We appreciate the reviewer for pointing out this.

According to the comment, we have deleted the "... is record-high in high-κ materials..." wording in Abstracted section and revised the "high-κ materials" to the "high-κ binary oxides", which is distinguished from the perovskite oxides, throughout the manuscript. In addition, we also revised Table S1, in which the materials are classified into two major groups, the high-κ binary oxides, including HfO₂-based and ZrO₂-based thin-film capacitors, and the perovskite oxides, covering paraelectric, (anti-)ferroelectric, and relaxor ferroelectric capacitors. In the main text, Fig. 4e is also revised by adding the data of all perovskite oxides, which makes the manuscript more preciseness and more general for a broad audience.

Related revisions can be found in line 7, page 2, line 6-11, page 4, line 17-18, page 4, and line 14-21, page 16 in the main text.

2) In the BHO02 thin film, the authors have described the coexistence of m- and o- phases. However, in the HAADF image, I only found the results of o-phase. The authors should provide the STEM results for the m-phase of the BHO02.

To comply with the reviewer, we have added the HAADF image of BHO02 sample with the coexistence of *m*- and *o*-phases, as shown in Fig. S5 in Supplementary Materials.

3) Also, in the structure characterizations in Fig. 1, I found that the amorphous regions of Ba-Hf-O are strongly non-stoichiometric. Noting that these samples have also been treated by the same annealing temperature for the crystalline HfO₂ and BaHfO₃. Why are there no secondary or impurity phases formed in

the structural evolution range?

It is indeed an interesting phenomenon in our samples. We also noticed this point when preparing the manuscript. We think that the absence of impurity phases might be ascribed to the maintaining of face-centered metal frames during the structure evolution since the structure similarity of the parent HfO_2 and BaHfO_3 . To evidence this hypothesis, we carried out an additional experiment by annealing the BHO12 sample with further elevating temperature, as shown in Fig. R1. One can find that the amorphous structure can persist to 1173 K, 200 K higher than the annealing temperature (973 K) adopted for preparing the *A*-Hf-O samples (see Methods section). When the temperature is increased to 1273 K (1373 K), the fluorite (perovskite) structure is crystallized. Until both the fluorite and perovskite structures are quenched at 1573 K, the impurity phases appear, as indicated by the yellow arrows. These phenomena may suggest that as long as the Hf/Ba metal frames persist, the impurity phases are hard to form in the Ba-Hf-O system even though it is strongly non-stoichiometric.

However, considering that the results in Fig. R1 is only a proof experiment for a possible effect of the structure-evolution strategy, these results aren't added to the manuscript.

Fig. R1. XRD patterns of the amorphous BHO12 thin film annealed by increasing temperature from 1073 K to 1573 K.

4) *In the amorphous structure, the metal-oxygen bonding may differ from that of the crystalline form. It is thus necessary to characterize the valence state of Hf and Ba ions.*

To comply with the reviewer, we have measured the valence state of Hf and Ba ions of the amorphous BHO12 sample by the means of XPS, as shown in Fig. R2. In the XPS spectrum of Hf 4f core level, the peaks at 16.3 and 18.0 eV are Hf 4f_{7/2} and Hf 4f_{5/2}, which are attributed to the Hf⁴⁺ of Hf-O bond. In the XPS spectrum of Ba 3d core level, the Ba 3d_{5/2} and 3d_{3/2} appear at 779.5 and 794.8 eV, respectively, separated by 15.3 eV, which correspond to the Ba²⁺ in Ba-O bond. There are no metallic Hf and Ba observed in the XPS measurements.

Fig. R2 has been added to the Supplementary materials as Fig. S8. Related revisions can be found in line 10-12, page 9 in the main text.

Fig. R2. XPS spectra of Hf 4f and Ba 3d core levels of the BHO12 thin film.

5) *In Fig. 3a, compared with the BHO02, I found that the hysteresis of BHO04 capacitor becomes weaker but this composition exhibits a more pronounced ferroelectric o-phase in the XRD patterns. Why?*

The suppression of hysteresis behavior in the BHO04 capacitor may be due to the formation of amorphous fractions since, according to the first-principles calculation, a low Ba concentration of 1/32 (~3.1%) can result in remarkable instability of the neighboring oxygen ions. This discussion has been added to the main text, as shown in line 4-9, page 13.

6) About the discussion of dielectric breakdown properties, it is better to measure leakage currents of the BHO capacitors as well, and include it in the paper.

Thank you for the suggestion. Current density-electric field curves of BHO capacitors have been measured, as shown in Fig. R3. One can find that the amorphous BHO12 capacitor exhibits the lowest leakage current, which is even close to the noise current of ~1.0 pA of our facilities.

Fig. R3 has been added to Supplementary Materials, as Fig. S12 and related discussion has been added to the main text, as shown in line 20-22, page 14.

Fig. R3. Current density-electric field curves of the BHO0, BHO02, BHO12, and BHO50 thin-film capacitors. Here, the noise of our facilities is also shown for comparison.

7) *It is also related to the discussion of breakdown. In this paragraph, the wording “Considering that the BHO thin films....., the electronic breakdown.....can be excluded” is inappropriate. In the BHO capacitors, the electronic breakdown cannot be excluded since it is an intrinsic character of dielectrics. In fact, what could be excluded in this case is that the electronic breakdown may be not a dominant factor for the difference in breakdown strength between the amorphous and crystalline BHO capacitors.*

Thank you very much for pointing out our mistake in the discussion. We have revised the discussion accordingly, as shown in line 3-6, page 14.

8) *Regarding the analysis of breakdown field with thickness, the negative power law used in Fig. S10 is a general description for breakdown behaviors of dielectrics. I suggest the authors employing specific formula of the avalanche mechanism to analyze the thickness dependent breakdown strength, such as the “40-generation-electron theory” mentioned in Ref. 1, which probably be better.*

Thank you for the suggestion. We have analyzed the thickness-dependent breakdown strength of the BHO0, BHO02, and BHO50 capacitors by the 40-generation-electron theory, as shown in Supplementary Fig. S12b. Related revision can be found in line 18-20, page 14 in the main text.

9) *For the calculation of formation of oxygen vacancy, there are oxygen rich and poor conditions. It seems like that the oxygen poor condition are considered. How does that calculation match the experimental environment?*

As the reviewer stated, for the chemical potential of oxygen, two cases are considered: oxygen rich and oxygen poor. The calculated $E^f(V_O)$ under oxygen rich condition is the upper limit while the $E^f(V_O)$ under oxygen poor condition is the lower limit. Fig. R4 demonstrates the $E^f(V_O)$ of the first nearest-neighbor of *t*-phase HfO₂ (as suggested by the reviewer #2) for both oxygen rich and poor conditions. In fact, the experimental environment of our work is between oxygen

poor and rich. Therefore, we provided the $E^f(V_O)$ under oxygen rich condition in the manuscript for ensuring the validity of the calculation results.

We have added the explanation of oxygen condition adopted in the first-principles calculation to the Methods section, as shown in line 1-2, page 19.

Fig. R4. $[E^f(V_O)]$ of t -HfO₂ at the first nearest-neighbor (NN) site as a function of the substitution concentration calculated in the oxygen rich and oxygen poor condition.

Finally, we thank the reviewer again for the invaluable suggestions and comments. The manuscript is indeed improved significantly after the revision. We hope this revised manuscript would meet the criteria for publication in Nature Communications.

Response to Reviewer #2

The manuscript by Yu et al. reported amorphous hafnium-based oxides for dielectric energy storage. They created amorphous hafnium-based oxides by bridging fluorite HfO₂ and perovskite hafnate. The structure evolution with different compositions was investigated by XRD and TEM. The formation mechanism of amorphization was tried to understand by DFT calculations. A breakdown field strength (E_b) of ~12 MV/cm and a recoverable energy storage density (U_{rec}) of ~155 J/cm³ were successfully obtained in this material system. This work proposed a structure-evolution strategy to overcome the negative correlation between E_b and permittivity. It may guide the research of energy storage materials and expand the application field of hafnia-based oxides, while some scientific problems should be solved.

We appreciate your time and effort for reviewing our manuscript, and cordially thank you for your comments that our work may guide the research of energy storage materials and expand the application field of hafnia-based oxides. All your comments have been addressed point-by-point as shown below and the manuscript has been revised correspondingly. The revisions are highlighted in blue in the revised manuscript.

1. In Table S1, the thickness in different samples is varied. What's the influence of the thickness to the structure evolution and energy storage performance?

According to the comment, we have deposited the Ba-Hf-O thin films with representative compositions of 4%, 6%, 12%, 15%, and 20% around the amorphous region in the thickness of 50 nm, as shown in Fig. R1. As shown, the 50 nm-thick BHO exhibits the same amorphous region of $4\% < x < 20\%$ with that of the 30 nm-thick BHO (Fig. 1c). The only difference is that the 50 nm-thick BHO04 film has more *m*-phase, as evidenced by the pronounced Bragg reflection at $2\theta = 34.2^\circ$. In addition, we also deposited a 10 nm-thick BHO12 thin film, which is also in the amorphous state, as shown in Fig. R2. Therefore, one may draw a conclusion that the film thickness has little effects on the structure evolution in our material

design strategy.

In Fig. R3, we measured the P - E loops of the amorphous BHO12 capacitors with different thicknesses of 10, 20, and 30 nm. One can find that these capacitors exhibit the same E_b of ~ 12 MV/cm and minor difference in the energy storage performance. In details, the leakage current of 10 nm-thick BHO12 is a bit larger than the 30 nm-thick sample, which results in a lower η of 70.4%. The increased leakage may be ascribed to the tunneling effect in the ultrathin insulating layer sandwiched between Pt and LSMO. For the 50 nm-thick BHO12 capacitor, the polarization is a bit smaller than that of the 30 nm-thick one, resulting in a slightly lowered U_{rec} of 122 J/cm³. Overall, the thickness effects on the energy storage properties aren't significant in our amorphous BHO films. Both the U_{rec} and η are varying within small ranges from 122 to 155 J/cm³ and 70 to 90% , respectively.

Regarding the previously reported energy storage performances in the high- κ binary oxides, as listed in Table S1, one may find a very large thickness change from 6 to 470 nm covering difference material systems from pure HfO₂ (ZrO₂), to doped HfO₂ (ZrO₂), and multilayerd films, in which the U_{rec} and η are also varying significantly from 18.17 to 109 J/cm³ and 50 to 94.4% , respectively. However, the origins for the performance differences among these thin-film capacitors are very complicated. A discussion on these results may be out of the scope of the present work.

Fig. R1. XRD patterns of the 50 nm-thick BHO thin films with increasing Ba concentration from 4% to 20%.

Fig. R2. XRD patterns of the 10, 30, and 50 nm-thick BHO12 thin films.

Fig. R3. *P-E* loops of the 10, 30, and 50 nm-thick BHO12 thin-film capacitors.

2. In Figure 1c, the diffraction peak at 30 degree was labeled as o-phase. But the diffraction peak of tetragonal (*t*-) phase is also around 30 degree. Why the authors exclude the possibility of *t*-phase in the film. Why the peak of BHO0 at 30 degree is not significant, but the peak at 28 degree is more significant than that in BHO02 and BHO04. What does the peak at 42 degrees stands for, it seems that the intensity gradually reduced from bottom (Fluorite HfO_2) to top (Perovskite BaHfO_3)?

Thank you for the comments. We agree with the reviewer that the t -phase may also coexist with the o -phase since the $(011)_t$ reflection is located at 30.05° and the 2θ difference between $(111)_o$ and $(011)_t$ is too small to be distinguished. We therefore added the discussion to the main text, as shown in line 14-16, page 7. We also revised Fig. 1c by adding the indication of t -phase at 2θ of $\sim 30^\circ$.

In Fig. 1c, the diffraction peak at 2θ of $\sim 30^\circ$ is o -phase HfO_2 (or with t -phase). The presence of o -phase is due to the Ba substitution-induced lattice distortion. Therefore, one can find the o -phase peaks in BHO02 and BHO04 thin films. For the BHO0, it is the undoped HfO_2 thin film, which is in the m -phase, the energetically-favorable phase of bulk HfO_2 , since there is no substitution-induced lattice strain. The Bragg reflection for m -phase HfO_2 is at 2θ of $\sim 28^\circ$. Therefore, one can find in the XRD patterns that the BHO0 exhibits a significant peak at $\sim 28^\circ$ while the BHO02 and BHO04 exhibit significant peaks at $\sim 30^\circ$. According to the comment, we are aware that the abbreviation BHO0 may be somewhat misleading. We therefore added an explanation of BHO0 when it first appears, as shown in line 10, page 7 in the main text.

The peak at 2θ of $\sim 42^\circ$ is k_β of STO substrate. We have indicated it in Fig. 1c and added a description in the caption.

3. The authors said that the amorphous BHO was formed in a high-temperature (973 K) crystallizing process and the HfO_2 should have a high symmetry, like the cubic (c -). Then, they adopted c -phase in the DFT calculation (Fig. 1g, h). Based on the phase diagram of HfO_2 , c -phase is an ultra-high temperature phase. At 973K, the HfO_2 should have a m -symmetry without dopants and pressure, or have a t -symmetry with dopants or pressure. The authors need to reevaluate the calculation result.

Thank you for the comments. We have carried out the first-principles calculation based on t -phase HfO_2 , which show similar results with that of the c -phase, as shown in Fig. 1g and 1h in the main text and Fig. S9 in Supplementary Materials. We think the similarity of the results is because t - and c -phases are similar in

structure.

Related revision can be found in line 10-16, page 8 and line 17-18, page 9 in the main text. The calculation results based on *c*-phase HfO₂ have been deleted from the manuscript.

4. In Figure 1e, small crystalline clusters in HAADF images and texture spot in FFT patterns can be found, did this suggest that the film is not in the amorphous state. Since the TEM image reflects a local area, more TEM evidences should be provided. Besides, the elemental distribution for Hf and Ba in Figure 1e seems also not very uniform, especially for Ba element as shown in Figure S5, is this the reason for the change in performance?

In the HAADF image of BHO12 (Fig. 1e), one can find there exist short-range ordering within local regions of several atoms. However, there are no long-range periodicities observed. These characters suggest the formation of amorphous state. The presence of texture spots on the diffraction ring in the FFT pattern may result from the short-range ordering within local regions of several atoms. We added these discussions to the main text, as shown in line 3-4, page 8.

Regarding the uniformity of the amorphous BHO film, we have provided additional HAADF image in a large area and EDS (energy dispersive spectra) measurements for the element distributions, as shown in Fig. R4a and R4c. In Fig. R4a, one can find that the BHO12 is in an amorphous state over a large area with the scale bar of 20 nm. The EDS mappings in Fig. R4c exhibit the uniform distributions of the Hf, Ba, Sr, La, and Mn elements. We also noticed the seemingly non-uniform in the EELS element mappings (Fig. R4b), which may be due to the limited step in the data collection. However, the EELS mappings are good at the characterization of stoichiometric difference between Hf and Ba. We therefore retain these results in the Figure.

Fig. R4 is also shown in Supplementary Materials, as Fig. S6. Related revision can be found in line 4-6, page 8 in the main text.

Fig. R4. STEM characterizations of the BHO12/LSMO/STO heterostructure in a large scale to show the uniformity in amorphous structure (a) and element distributions (b, c). b, the EELS mappings and c, the EDS mappings.

5. In Figure 1g, h and Figure S7, why did the authors only choose the c-phase, m-phase and o-phase in the DFT calculations. The t-phase should also be considered.

As suggested by the reviewer, we have provided the first-principles calculation results based on t-phase HfO_2 , as shown in Fig. 1g and 1h in the main text and Fig. S9 in Supplementary Materials.

6. The two fitting curves for the EXAFS data do not match the experimental results very well, especially in the upper curves.

In the fitting of EXAFS data, the fitting window is set to $R = 1.0 \sim 4.0 \text{ \AA}$, which is typical for the EXAFS analysis of amorphous structure because (i) $R < 1.0 \text{ \AA}$ is meaningless since there is no interatomic distance shorter than 1.0 \AA in actual crystals; (ii) the oscillation amplitude of EXAFS spectrum is too low to be fitted when $R > 4.0 \text{ \AA}$ since the amorphous structure only has short-range ordering. Within the fitting window, the fitting of EXAFS data in Fig. 2 is good with very low R-factor of 0.0065 for the BHO12-RT (the inset) and 0.0092 for the BHO12. To make the representation more clearly, we have added the indication about the fitting window in Fig. 2. The explanation for the set of fitting window is also added to Supplementary Text 1.

7. In the last paragraph of "Introduction", the authors took a lot of words to describe the crystal structure of HfO_2 and AHfO_3 and the strategy of the amorphous structure designing in Fig. 1a. These contents seem inappropriate to be described in "Introduction", but in the first part of "Results". Besides, there were many detailed and conclusive sentences about the physical mechanisms. However, these conclusions can only be reached after describing the experimental results and thus should be moved to the "Conclusions" or "Discussions"

As suggested by the reviewer, we have revised the last paragraph of the Introduction section, as shown in line 12-18, page 4. In this paragraph, only the major results and conclusions of the present work are retained. The discussion on Fig. 1a has been moved to the Results section as the first paragraph of the "Amorphization of hafnium-based oxides" subsection, in which we also revised some conclusive sentences, as shown in line 12-24, page 6. These revisions make the organization more reasonable.

Finally, we thank the reviewer again for the invaluable suggestions and comments. The manuscript is indeed improved significantly after the revision. We hope this revised manuscript would meet the criteria for publication in Nature Communications.

REVIEWERS' COMMENTS

Reviewer #1 (Remarks to the Author):

The authors addressed all of the questions. I recommend the present paper for publication.

Reviewer #2 (Remarks to the Author):

My concerns have been addressed. I can now recommend the publication of this manuscript in Nature Communications.

Response to Reviewer #1

The authors addressed all of the questions. I recommend the present paper for publication.

Thank you very much for your approval on our revisions!

It is very encouraging for us to publish this work on Nature Communications, which is indeed a good starting for our research on the structure evolution and its new functionality between different material systems. We also would like to express our sincere gratitude to your invaluable suggestions and comments in the review.

Response to Reviewer #2

My concerns have been addressed. I can now recommend the publication of this manuscript in Nature Communications.

Thank you very much for your approval on our revisions!

It is very encouraging for us to publish this work on Nature Communications, which is indeed a good starting for our research on the structure evolution and its new functionality between different material systems. We also would like to express our sincere gratitude to your invaluable suggestions and comments in the review.